# TELeR: A General Taxonomy of LLM Prompts for Benchmarking Complex Tasks

**Shubhra Kanti Karmaker ("Santu"), Dongji Feng**
Big Data Intelligence (BDI) Lab
Department of Computer Science & Software Engineering
Auburn University, Alabama, USA
{sks0086, dzf0023}@auburn.edu

## Abstract

While LLMs have shown great success in understanding and generating text in traditional conversational settings, their potential for performing ill-defined complex tasks is largely under-studied and yet to be benchmarked. However, conducting such benchmarking studies is challenging because of the large variations in LLMs' performance when different prompt types/styles are used and different degrees of detail are provided in the prompts. To address this issue, this paper proposes a general taxonomy that can be used to design prompts with specific properties in order to perform a wide range of complex tasks. This taxonomy will allow future benchmarking studies to report the specific categories of prompts used as part of the study, enabling meaningful comparisons across different studies. Also, by establishing a common standard through this taxonomy, researchers will be able to draw more accurate conclusions about LLMs' performance on a specific complex task.

## 1 Introduction

Recently, conversational Large Language Models (LLMs) such as GPT-3 (Brown et al., 2020), Bard (Thoppilan et al., 2022), LLaMA (Touvron et al., 2023), BLOOM (Scao et al., 2022), PaLM (Chowdhery et al., 2022), etc. have demonstrated exceptional performance in a wide range of popular natural language processing (NLP) tasks (Bubeck et al., 2023; Dai et al., 2022; Du et al., 2022; Smith et al., 2022). Prompt, as a stimulator, refers to a textual input provided to the LLMs with the intention of guiding its output toward a specific task. Unsurprisingly, the quality and effectiveness of the prompt can greatly influence the performance of the LLMs for a particular task, and therefore, designing appropriate prompts with the right amount of detail has become more important than ever (Liu et al., 2023; Han et al., 2022).

In recent years, researchers have spent a significant amount of effort proposing different ways of designing "appropriate" prompts. For example, Brown et al. (2020) showed a standard prompting technique with question-answer pairs that can result in a few-shot effect. Researchers also explored other prompt design techniques such as Chain-of-thought (CoT) (Wei et al., 2022), Reasoning and Acting (ReAct) (Yao et al., 2022), and other techniques (Kojima et al., 2022; Madaan and Yazdanbakhsh, 2022; Press et al., 2022) in terms of improving the reasoning and acting of LLMs in solving Question-Answering tasks. Meanwhile, Kim et al. (2023) proposed a prompting scheme where the agent recursively criticizes and improves its output (RCI) to solve a task. However, these experiments primarily emphasized the utilization of diverse prompts to evaluate the ability of LLMs to perform "well-defined" NLP tasks, while studies with diverse prompts for ill-defined complex tasks are still rare, if not nonexistent.

While conducting multiple benchmarking studies with various LLMs for complex tasks seems interesting and compelling, conducting such studies is challenging because of the large variations in LLMs' performance when different prompt types/styles are used and different degrees of detail are provided in the prompts, especially in case of complex tasks. In this paper, we exclusively focus on understanding LLMs' potential for performing complex tasks that are mostly: 1) ill-defined, 2) abstract goal-oriented, 3) highly dependent on subjective interpretation, and 4) very hard to evaluate quantitatively (Khot et al., 2022; Press et al., 2022). These complex tasks often involve multiple steps/sub-tasks, and designing "appropriate" prompts for such tasks is indeed challenging as there is no single rule book to follow in these cases (Zelikman et al., 2022; Nye et al., 2021). A further complication arises if we want to compare two independent benchmarking studies targeted

towards the same goal (complex) task. Such a complication arises because, for a given complex task and a particular LLM, the performance of the LLM can drastically vary when different types/styles of prompts are fed to it. Indeed, the exact details included in the prompt play a big role in how LLMs will perform in solving the goal complex task. This indeed creates a problem for evaluation and benchmarking purposes if an apple-to-apple comparison is not made in terms of the prompts that are provided to the LLMs. In other words, just reporting accuracy numbers for LLMs without specifying the finer details of the prompts used in the experiments makes comparisons across LLMs meaningless.

Unfortunately, every complex task is different, and so are the prompts users can try to perform the task; therefore, a general taxonomy that can categorize these diverse kinds of prompts using a single standard/taxonomy has now become a pressing need. The main contribution of this paper is to introduce one such general taxonomy (we name it TELeR) that can be used by any benchmarking study that leverages LLMs to perform some complex task. The major benefit of adopting our proposed TELeR taxonomy is that it will facilitate more meaningful comparisons among multiple LLMs in terms of their performances across various complex tasks reported by multiple independent groups of researchers/developers and, thus, help derive more accurate conclusions. TELeR will achieve this goal by grounding different types of prompts into a common standard and allowing an apple-to-apple comparison across different prompt categories using the same standard. As such, this taxonomy will serve as an important tool to establish a common consensus around the state-of-the-art LLM performance for performing complex tasks.

## 2 Prompt Engineering for Complex Tasks

"Prompt Engineering" is a crucial technique for maximizing the utility of LLMs in various tasks (Zhou et al., 2022). It involves crafting and revising the query or context in such a way that it elicits the desired response or behavior from LLMs (Brown et al., 2022). In practice, prompt engineering is an iterative process requiring multiple trial and error runs (Shao et al., 2023).

*Prompt Engineering* becomes even more critical and challenging in case of performing complex tasks (Tan et al., 2023) with LLMs, as complex tasks usually involve multiple steps/sub-tasks requiring higher levels of semantic understanding, planning, reasoning, and generation of natural language (Fu et al., 2022). These tasks often require the model to go beyond simple pattern recognition or retrieval of information and involve simulating more advanced cognitive abilities. In fact, differences in prompts along several key factors can have a significant impact on the accuracy and performance of LLMs in complex tasks. Below, we list those key factors of prompt designing.

- **Level of Details in Task Specification**: The prompt directive should define the task or question being asked in sufficient detail (White et al., 2023; Ouyang et al., 2022). For complex tasks, providing a detailed directive typically means following the general guidelines below.

  – **Clear Goal(s):** Specifying clear goals helps guide the language model's understanding of the task or question at hand, increasing the chances of receiving the desired information or output. Therefore, one should avoid vague or ambiguous terms that can lead to inaccurate or irrelevant responses (Jiang et al., 2022).

  – **Associated Data:** Some prompts require LLMs to perform a particular task on the data provided by the user in real-time, whereas some prompts do not provide any data and rely on the pre-trained model to generate a response based on the background knowledge it has already learned. It is very important to make it explicit in LLM prompts whether the user is providing data as part of the prompt or not, and if yes, which part is data vs. directive.

  – **Distinct Sub-Tasks:** By definition, complex tasks consist of multiple steps/ sub-tasks. It is important to mention these distinct sub-tasks in the prompt clearly as separate bullet points or numbered items. This visual organization helps LLMs recognize the distinct sub-tasks and respond to each one individually.

  – **Evaluation Criteria/Few-Shot Examples:** LLMs can benefit from example-based learning, where prompts include specific examples of the desired input-output pairs (few-shot examples) (Brown et al., 2020). By incorporating relevant examples, users can train the model to follow specific patterns or mimic desired behavior. In the absence of explicit few-shot examples, prompts may describe what consti-

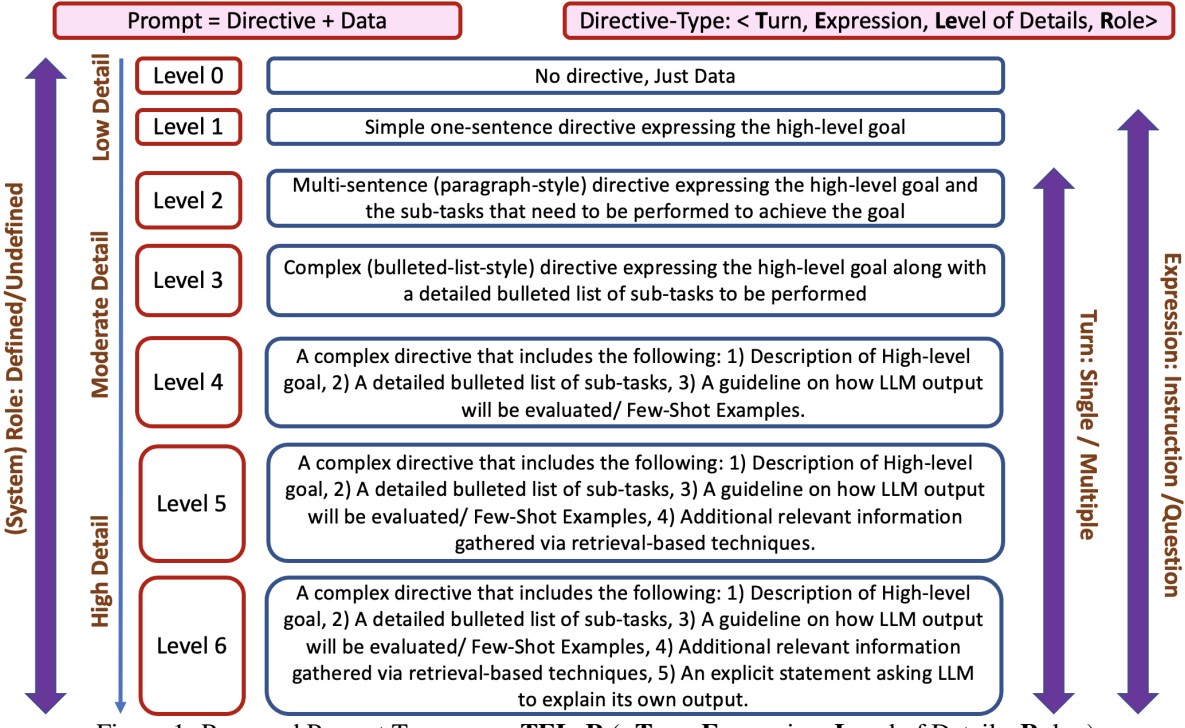

Figure 1: Proposed Prompt Taxonomy: **TELeR** (<**T**urn, **E**xpression, **Le**vel of Details, **R**ole>)

tutes a "good" response versus what would make a response "bad". For example, imposing word limits, specifying output formats, or restricting particular data sources etc.

– **Additional Information Fetched via Information Retrieval Techniques:** Additional information fetched via Information Retrieval Techniques enhances Large Language Models (LLMs) by providing them with real-time and contextually relevant data, improving their ability to generate up-to-date and accurate responses. This helps LLMs stay current and adapt to evolving information, making them more valuable for various applications, such as chatbots and search engines.

– **Explanation/Justification Seeking:** LLMs are not only good at generating textual responses, but they can also generate explanations for their output if an explanation is sought as part of the prompt explicitly (Rajani et al., 2019). This is indeed valuable when a user wants to understand why the LLM generated a particular output.

• **Defining Context and Role**: Including relevant context and background information as part of the prompt can provide the model with complementary information in order to generate more accurate responses. For complex tasks, giving the model a clear understanding of the context

can help it make more informed and precise decisions. Different prompts may provide varying levels of context, which can impact the accuracy of the model's responses.

• **Expression Style**: Directives can be expressed primarily in two styles: 1) Questions and 2) Instructions. For complex tasks, one may frame directives as either a set of questions or instructions based on their preference/application need.

• **Interaction Style**: Prompts for complex tasks usually involve long text descriptions, often including details of associated sub-tasks to be performed step-by-step. Therefore, some users may prefer to provide these step-by-step instructions in a multi-turn fashion (like a real dialog), whereas others may prefer to provide all the details at a single turn. Such one-turn vs. multi-turn prompting can also impact the performance of an LLM significantly as the dialog history becomes different in generation time for these two cases.

## 3 Proposed TELeR Taxonomy

Our proposed taxonomy is based on the key factors we discussed in section 2, whose variations can lead to different outcomes while using LLMs to perform complex tasks. To begin with, we represent each prompt as a combination of *Directive* and *Data*. Assuming *Data* is fixed for a given goal task, the difference between two prompts essen-

tially originates from the specifics/details of directives they include. More specifically, we propose to categorize LLM prompts for complex tasks along the following four dimensions.

1. **Turn:** Based on the number of turns used while prompting LLMs in order to perform a complex task, prompts can be either single or multi-turn.

2. **Expression:** Based on the expression style of the overall directive as well as the associated sub-tasks, prompts can be either question-style or instruction-style.

3. **Role:** Based on whether a proper system role is defined in the LLM system before providing the actual prompt, prompts can be categorized as either system-role *defined* or *undefined*.

4. **Level of Details:** Based on the *degree of detail* provided in the directive, we divided prompts into seven distinct levels (levels 0-6). Here, the degree of detail is determined by the presence or absence of different aspects like clear goals, sub-task division, explanation seeking, few-shot examples, etc. By definition, Level "0" means minimal details, i.e., no aspects/no directive, while Level "6" means the highest level of details where the directive includes clear goals, distinct sub-tasks/steps, an explicit requirement of explanation/justification, well-defined criteria for evaluation, additional information fetched via information retrieval techniques and/or few-shot examples. See Figure 1 for the exact definitions of each of these levels in our taxonomy.

Because we used the following four factors, i.e., *Turn*, *Expression*, *Level of Details* and *Role*, to define our taxonomy, we name it as **TELeR**. The overall taxonomy is pictorially depicted in Figure 1.

## 4 Two Example Use Cases

In this section, we present two interesting use cases of the proposed TELeR taxonomy that involve LLMs for performing a complex task: 1) generating a meta-review from peer-reviewer comments on a scholarly work, and 2) Combining multiple alternative narratives into a single braided one.

### 4.1 Use-Case 1: Meta-Review Generation

Meta-reviewing is a critical part of the scientific peer-review process and is generally a complex task that involves summarizing expert reviews from multiple reviewers (Shen et al., 2022, 2023). It is a very important and pertinent process for making informed decisions and understanding the consensus of expert opinions on a submitted manuscript. Given the explosion in the number of research manuscript submissions in recent years and the huge challenge in peer-reviewing timeline management (Bansal et al., 2022c; Karmaker Santu et al., 2018), it is tempting to leverage LLMs to assist editors (for journals)/ program chairs (for conferences) in preparing a first-cut draft of the meta-review for each manuscript from the individual review texts provided by relevant expert reviewers.

To demonstrate the applicability of the proposed TELeR taxonomy for categorizing different kinds of prompts for this complex task, we show some example prompts with varying levels of detail below. For simplicity, we show examples of only single-turn question-style prompts where the system role is undefined. Other variations are left out due to lack of space. We also assume that three reviewers have reviewed the manuscript and provided their comments ($R_1$, $R_2$, $R_3$) as the *data* for the meta-review generation task.

- **Level 0 Prompt:** <$R_1$, $R_2$, $R_3$>
- **Level 1 Prompt:** *Prepare a meta-review by summarizing the reviewer comments:* <$R_1$, $R_2$, $R_3$>
- **Level 2 Prompt:** *Prepare a meta-review by summarizing the following reviewer comments. The final output should highlight the core contributions of the manuscript, common strengths/weaknesses mentioned by multiple reviewers, suggestions for improvement, and missing references (if any). The review texts are provided below:* <$R_1$, $R_2$, $R_3$>
- **Level 3 Prompt:** *Prepare a meta-review by answering the following questions from the reviewer comments (provided after the questions).*

  1. *Based on the reviewer's comments, what are the core contributions made by the authors?*
  2. *What are the common strengths of this work, as mentioned by multiple reviewers?*
  3. *What are the common weaknesses of this work, as highlighted by multiple reviewers?*
  4. *What suggestions would you provide for improving this paper?*
  5. *What are the missing references mentioned by the individual reviews?*

  *The review texts are below:* <$R_1$, $R_2$, $R_3$>

- **Level 4 Prompt:** "Level 3 Prompt" + *"A good output should be coherent, highlight major strengths/issues mentioned by multiple reviewers,*

*be less than 400 words in length, and finally, the response should be in English only".*

- **Level 5 Prompt:** "Level 4 Prompt" + *"Below are additional information relevant to your goal task. <Information Fetched using Information Retrieval Techniques>".*
- **Level 6 Prompt:** "Level 5 Prompt" + *"Justify your response in detail by explaining why you made the choices you actually made".*

## 4.2 Use Case 2: Narrative Braiding

Narrative braiding, also known as "interweaving" or "multi-perspective storytelling" is a literary technique that involves the parallel telling of multiple storylines that eventually converge and intersect (Bancroft, 2018). This technique is often used in novels, short stories, films, and television shows to create a complex and engaging narrative.

Narrative braiding is indeed a complex task to perform, even for humans, let alone computers, as it requires careful planning and execution to ensure that each storyline is fully developed and that the different strands of the narrative are balanced and complement each other. When done well, narrative braiding can create a rich and engaging story that keeps readers or viewers invested. From the recent promising results of language models in generating high-quality controlled text (Bansal et al., 2022a,b), it is quite intuitive to test the LLM's performance in narrative braiding tasks.

Now, we show how one can use the proposed TELeR taxonomy to categorize different types of prompts for the narrative braiding task. This time, we show examples of only single-turn instruction-style prompts with system roles undefined. Other variations are left out due to lack of space. We also assume two alternative narratives are available that describe the same event as our *data* for the braiding task, and the goal is to create a final braided narrative from the two input narratives, $N_1$ and $N_2$.

- **Level 0:** <$N_1$, $N_2$>
- **Level 1:** *Braid a single coherent story from the following alternative narratives:* <$N_1$, $N_2$>
- **Level 2:** *Braid a single coherent story from the following alternative narratives. The final narrative should highlight the common information provided by both narratives, interesting, unique information provided by each individual narrative, and conflicting information (if any) conveyed in these narratives. The input alternative narratives are provided below:* <$N_1$, $N_2$>

- **Level 3:** *Braid a single coherent story from the following alternative narratives provided later by performing the following tasks.*
  1. *Extract overlapping clause pairs from both narratives and paraphrase them.*
  2. *Extract unique clauses from each narrative and identify the interesting ones.*
  3. *Extract conflicting clause pairs conveyed in both narratives and resolve the conflict.*
  4. *Generate paragraphs from overlapping-unique-conflicting clauses and merge them into a single document.*
  5. *Reorder sentences of the merged document into a detailed, coherent story.*
  6. *Summarize the detailed story into a concise, braided narrative.*

  *The alternative narratives are below:* <$N_1$, $N_2$>
- **Level 4 Prompt:** "Level 3 Prompt" + *"A good output should be coherent, highlight overlapping-unique-conflicting information provided by individual narratives, be less than 1000 words in length, and in English language only".*
- **Level 5 Prompt:** "Level 4 Prompt" + *"Below are additional information relevant to your goal task. <Information Fetched using Information Retrieval Techniques>".*
- **Level 6 Prompt:** "Level 5 Prompt" + *"Provide justification for your response in detail by explaining why your response contains certain information and discards other information of the inputs".*

## 5 Final Words

In this paper, we emphasize the importance of a standardized taxonomy for LLM prompts targeted towards solving complex tasks and, subsequently, propose such a general taxonomy, i.e., TELeR, which can serve as a unified standard for comparing and benchmarking LLMs' performances reported by multiple independent research studies. We urge the community to use the TELeR taxonomy for designing prompts in their future work and report the specific categories of prompts they experimented with in their manuscripts. This will enable more meaningful comparisons among LLMs and, thereby, help to derive more accurate conclusions from multiple independent studies. This, in turn, will help the community to reach a consensus on state-of-the-art LLM performances more accurately and faster than otherwise.

# 6 Limitations

The proposed TeLER taxonomy is exclusively applicable to LLM prompts targeted toward solving a complex task. This taxonomy, especially the seven levels, does not apply to simple tasks. As such, TeLER taxonomy will be most useful to researchers and developers who are conducting applied LLM research and development that is focused on performing complex tasks.

Also, the TeLER taxonomy should not be considered as an ultimate taxonomy for LLM prompts, and further extensions of this taxonomy are certainly possible and actually desired. Having said that, the TeLER taxonomy is actually very general and can be easily extended by adding more dimensions for categorization as deemed necessary by the target application.

# 7 Acknowledgements

This work has been partially supported by the National Science Foundation (NSF) Standard Grant Award #2302974 and Air Force Office of Scientific Research Grant/Cooperative Agreement Award #FA9550-23-1-0426. We would also like to thank Auburn University College of Engineering and the Department of CSSE for their continuous support through Student Fellowships and Faculty Startup Grants.

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
