# OpenReview forum: "TELeR: A General Taxonomy of LLM Prompts for Benchmarking Complex Tasks"
_EMNLP/2023/Conference — EMNLP 2023 Findings_

### Official Review · Reviewer_SYHs · 2023-08-04

**Soundness:** 3

**Excitement:**

2: Mediocre: This paper makes marginal contributions (vs non-contemporaneous work), so I would rather not see it in the conference.

**Paper Topic And Main Contributions:**

To address the challenge of varying LLM performance with different prompt types and details, the authors provide a general taxonomy to design prompts with specific properties for various tasks. Based on this taxonomy, future benchmarking studies may be able to draw more accurate conclusions about LLMs' performance on specific complex tasks.

**Reasons To Accept:**

1. This paper is well-organized and easy to follow.
2. Examples of using this taxonomy for prompt categorizing are provided for better understanding.


**Reasons To Reject:**

1. The authors concentrate on presenting a taxonomy of prompts used in LLMs, but there is a need for a more comprehensive analysis of each category. For example, providing suggestions on which kinds of prompts are suitable for particular tasks or models will be meaningful and beneficial for future research.
2. To achieve a comprehensive understanding of the differences of these classes, the authors should include quantitative comparisons using various datasets and baselines. Considering this, this paper is better suited as a long paper.


**Reproducibility:**

N/A: Doesn't apply, since the paper does not include empirical results.

**Reviewer Confidence:**

3: Pretty sure, but there's a chance I missed something. Although I have a good feel for this area in general, I did not carefully check the paper's details, e.g., the math, experimental design, or novelty.

---

> ### Author Rebuttal · Authors · 2023-08-26
>
> Thank you for your comment. We want to highlight that this is a short theme track "position" paper. Our "position" here is that the current practice of deploying prompt engineering techniques in LLMs without a structured taxonomy is problematic, and we need a standard prompting taxonomy to enable a fair comparison of LLMs and, thus, derive more accurate and meaningful conclusions from different independent studies. We agree that it would be great to see if the taxonomy is really helpful by creating prompts and feeding them to LLMs with results reported. However, a 4-page limit is impractical to report such studies with details of different instances of prompts created. In fact, we have already used this taxonomy in multiple use cases and found it to be applicable to a wide range of complex tasks. In our paper, we have demonstrated the applicability of the TeLER taxonomy to two complex tasks, i.e., Meta-Review Generation (Section 4) and Narrative Braiding (Appendix). TeLER taxonomy can also be applied to other complex tasks like Comparative Analysis of Privacy Policy, Compiler Fuzzing, Social Media Bias Analysis, etc. We also found that Levels 3 and 4 are often performing better than other levels. However, we cannot point out/cite those studies as it will break the anonymity rules.

---

### Official Review · Reviewer_YfVr · 2023-08-05

**Soundness:** 3

**Excitement:**

2: Mediocre: This paper makes marginal contributions (vs non-contemporaneous work), so I would rather not see it in the conference.

**Paper Topic And Main Contributions:**

The paper presents a general taxonomy for designing prompts, called TELeR, for querying large language models (LLMs) on complex tasks.
The main contributions include enabling meaningful comparisons across different benchmarking studies, establishing a common standard, and facilitating more accurate conclusions about LLMs' performance on specific complex tasks.



**Questions For The Authors:**

1. Do all the tasks (or prompts) fall within the scope of this taxonomy? I believe that certain tasks which have specific requirements and evaluation procedures may not be practical to use, such as machine translations or reading comprehension. I am curious about the extent to which this taxonomy can be implemented.

**Reasons To Accept:**

1. Clarifying instructions is a crucial task, yet there is currently no standardized method for assessing their clarity. This paper touched on this important topic and proposed a taxonomy.

2. The facets under consideration are thorough, exhibiting a strong resonance with the intuitions and experiences of querying LLM.


**Reasons To Reject:**

1. There are some questions related to the application of this taxonomy that have not been addressed. When designing a taxonomy, it's essential to consider how prompts will be categorized and how they can be transformed to different levels. These questions should be addressed systematically, besides case studies for a specific task.

2. The performance gaps of models when handling queries of different prompt levels are also worth investigating. One may naturally consider the more detailed a prompt, the better the performance model achieves. However, does this hold for all tasks and different models (closed and open-source) is unclear.

3. The paper writing needs improving, and sometimes the content is not easy to follow. For example, the term 'complex task' is somehow ambiguous. Formally defining what kind of complex is focused on (e.g., a task requires in-depth knowledge, or includes a long horizon of planning) is highly preferred.

**Reproducibility:**

N/A: Doesn't apply, since the paper does not include empirical results.

**Reviewer Confidence:**

4: Quite sure. I tried to check the important points carefully. It's unlikely, though conceivable, that I missed something that should affect my ratings.

---

> ### Author Rebuttal · Authors · 2023-08-26
>
> - "There are some questions related to the application of this taxonomy that have not been addressed. When designing a taxonomy, it's essential to consider how prompts will be categorized and how they can be transformed to different levels. These questions should be addressed systematically, besides case studies for a specific task."  "The performance gaps of models when handling queries of different prompt levels are also worth investigating. One may naturally consider the more detailed a prompt, the better the performance model achieves. However, does this hold for all tasks and different models (closed and open-source) is unclear."
>
>   - Thank you for your comment. We want to highlight that this is a short theme track "position" paper. Our "position" here is that the current practice of deploying prompt engineering techniques in LLMs without a structured taxonomy is problematic, and we need a standard prompting taxonomy to enable a fair comparison of LLMs and, thus, derive more accurate and meaningful conclusions from different independent studies. We agree that it would be great to see if the taxonomy is really helpful by creating prompts and feeding them to LLMs with results reported. However, a 4-page limit is impractical to report such studies with details of different instances of prompts created. In fact, we have already used this taxonomy in multiple use cases and found it to be applicable to a wide range of complex tasks. We also found that Levels 3 and 4 are often performing better than other levels. However, we cannot point out/cite those studies as it will break the anonymity rules.
>
>
> - The paper writing needs improving, and sometimes the content is not easy to follow. For example, the term 'complex task' is somehow ambiguous. Formally defining what kind of complex is focused on (e.g., a task requires in-depth knowledge, or includes a long horizon of planning) is highly preferred.
>
>   - Thank you for your comment. The proposed TeLER taxonomy is actually very general and can be applied to any "complex" task. Our definition of a "complex" task is that the goal task must consist of multiple sub-tasks that need to be accomplished to reach the final output. The sub-tasks may be sequential or parallel in nature or a combination of both. In our paper, we have demonstrated the applicability of the TeLER taxonomy to two complex tasks, i.e., Meta-Review Generation (Section 4) and Narrative Braiding (Appendix). TeLER taxonomy can also be applied to other complex tasks like Comparative Analysis of Privacy Policy, Compiler Fuzzing, Social Media Bias Analysis, etc. Finally, the TeLER taxonomy can be easily extended by adding more dimensions for categorization as deemed necessary by the target application, and, therefore, the recall of covered tasks can improve as well allowing maximal generality.
>
>
>
> - "Do all the tasks (or prompts) fall within the scope of this taxonomy? I believe that certain tasks which have specific requirements and evaluation procedures may not be practical to use, such as machine translations or reading comprehension. I am curious about the extent to which this taxonomy can be implemented."
>
>   - The proposed TeLER taxonomy is exclusively applicable to LLM prompts targeted toward solving a complex task. This taxonomy, especially the six levels, does not apply to simple tasks like sentiment classification or sentence-level inference tasks. As such, TeLER taxonomy will be most useful to researchers and developers who are conducting applied LLM research and development that is focused on performing complex tasks that are hard to evaluate. Also, the TeLER taxonomy should not be considered as an ultimate taxonomy for LLM prompts, and further extensions of this taxonomy are certainly possible and actually desired. Having said that, the TeLER taxonomy is actually very general and can be easily extended by adding more dimensions for categorization as deemed necessary by the target application.

---

### Official Review · Reviewer_gD5n · 2023-08-05

**Soundness:** 3

**Excitement:**

2: Mediocre: This paper makes marginal contributions (vs non-contemporaneous work), so I would rather not see it in the conference.

**Paper Topic And Main Contributions:**

This paper proposed a Taxonomy for LLM prompt for complex tasks with the intent to provide a standard to evaluate the performance of LLMs, with the hope to remove the difference caused by prompt designs.

The proposed Taxonomy defines Prompt as Directive + Data, with 6 levels of details, and turn type, instruction vs question, and defined vs undefined roles.

The paper gave an example of how to design prompts to write Meta-reviews based on the proposed Taxonomy to show the usefulness of the proposal.

**Reasons To Accept:**

1. The idea itself is interesting and I believe it would be useful when we need to compare LLM performances. It can help the published LLM performance numbers are using the level of prompts, so that it's easier to compare
2. The Taxonomy seems to reasonable to me with enough elements, so it could be useful.

**Reasons To Reject:**

1. It only showed the proposed Taxonomy. I wanted to see some sort of examples showing how the TELeR really helped evaluate performance across the LLMs with a complex question. The given example is about generating prompts. It would be great to see it is really helped by feeding these to LLMs with results reported.
2. The coverage of proposed frameworks. It would be great to see some types of recall numbers of the proposed framework. Even though this paper admitted it only intended to cover complex problems, I wanted to see what types of complex problems the proposed framework can cover.

**Reproducibility:**

N/A: Doesn't apply, since the paper does not include empirical results.

**Reviewer Confidence:**

4: Quite sure. I tried to check the important points carefully. It's unlikely, though conceivable, that I missed something that should affect my ratings.

---

> ### Author Rebuttal · Authors · 2023-08-26
>
> - "It only showed the proposed Taxonomy. I wanted to see some sort of examples showing how the TELeR really helped evaluate performance across the LLMs with a complex question. The given example is about generating prompts. It would be great to see it is really helped by feeding these to LLMs with results reported."
>
>   - Thank you for your comment. We want to highlight that this is a short theme track "position" paper. Our "position" here is that the current practice of deploying prompt engineering techniques in LLMs without a structured taxonomy is problematic, and we need a standard prompting taxonomy to enable a fair comparison of LLMs and, thus, derive more accurate and meaningful conclusions from different independent studies. We agree that it would be great to see if the taxonomy is really helpful by creating prompts and feeding them to LLMs with results reported. However, a 4-page limit is impractical to report such studies with details of different instances of prompts created.  In fact, we have already used this taxonomy in multiple use cases and found it to be applicable to a wide range of complex tasks. We also found that Levels 3 and 4 are often performing better than other levels. However, we cannot point out/cite those studies as it will break the anonymity rules.
>
> - "The coverage of proposed frameworks. It would be great to see some types of recall numbers of the proposed framework. Even though this paper admitted it only intended to cover complex problems, I wanted to see what types of complex problems the proposed framework can cover."
>
>   - The proposed TeLER taxonomy is actually very general and can be applied to any "complex" task. Our definition of a "complex" task is that the goal task must consist of multiple sub-tasks that need to be accomplished to reach the final output. The sub-tasks may be sequential or parallel in nature or a combination of both. In our paper, we have demonstrated the applicability of the TeLER taxonomy to two complex tasks, i.e., Meta-Review Generation (Section 4) and Narrative Braiding (Appendix). TeLER taxonomy can also be applied to other complex tasks like Comparative Analysis of Privacy Policy, Compiler Fuzzing, Social Media Bias Analysis, etc. Finally, the TeLER taxonomy can be easily extended by adding more dimensions for categorization as deemed necessary by the target application, and, therefore, the recall of covered tasks can improve as well, allowing maximal generality.

---

### Meta-Review · Area_Chair_cPqU · 2023-09-17

**Recommendation:** 3

**Metareview:**

This paper presents a taxonomy that can be used to design prompts for LLMs to perform "complex" tasks. The authors describe the taxonomy and argue for it's usefulness, including a use case of meta-review generation. There are some concerns about the definition of a complex task, which is somewhat ambiguous in the paper, which have been answered in the rebuttal. Furthermore, there are some concerns about the practical usefulness of the taxonomy, which the authors assert is useful but it cannot be proven due to anonymity, meaning that this cannot be considered for the purposes of the review.

---

### Decision · Program_Chairs · 2023-10-07

**Decision:**

Accept-Findings

**Comment:**

This paper presents a taxonomy that can be used to design prompts for LLMs to perform "complex" tasks. The authors describe the taxonomy and argue for it's usefulness, including a use case of meta-review generation. There are some concerns about the definition of a complex task, which is somewhat ambiguous in the paper, which have been answered in the rebuttal. Furthermore, there are some concerns about the practical usefulness of the taxonomy, which the authors assert is useful but it cannot be proven due to anonymity, meaning that this cannot be considered for the purposes of the review.